# Bioelectronic Nose Based on Single-Stranded DNA and Single-Walled Carbon Nanotube to Identify a Major Plant Volatile Organic Compound (p-Ethylphenol) Released by Phytophthora Cactorum Infected Strawberries

**DOI:** 10.3390/nano10030479

**Published:** 2020-03-07

**Authors:** Hui Wang, Yue Wang, Xiaopeng Hou, Benhai Xiong

**Affiliations:** 1State Key Laboratory of Animal Nutrition, Institute of Animal Sciences, Chinese Academy of Agricultural Sciences, Beijing 100193, China; wangyue9313@163.com; 2Research Institute of Wood Industry, Chinese Academy of Forestry, Beijing 100091, China; houxiaopeng_lunwen@163.com

**Keywords:** bioelectronic nose, FET, ssDNA, SWNT, gas sensor, *Phytophthora cactorum*, volatile organic compounds, strawberry

## Abstract

The metabolic activity in plants or fruits is associated with volatile organic compounds (VOCs), which can help identify the different diseases. P-ethylphenol has been demonstrated as one of the most important VOCs released by the Phytophthora cactorum (*P. cactorum*) infected strawberries. In this study, a bioelectronic nose based on a gas biosensor array and signal processing model was developed for the noninvasive diagnostics of the *P. cactorum* infected strawberries, which could overcome the limitations of the traditional spectral analysis methods. The gas biosensor array was fabricated using the single-wall carbon nanotubes (SWNTs) immobilized on the surface of field-effect transistor, and then non-covalently functionalized with different single-strand DNAs (ssDNA) through π–π interaction. The characteristics of ssDNA-SWNTs were investigated using scanning electron microscope, atomic force microscopy, Raman, UV spectroscopy, and electrical measurements, indicating that ssDNA-SWNTs revealed excellent stability and repeatability. By comparing the responses of different ssDNA-SWNTs, the sensitivity to P-ethylphenol was significantly higher for the s6DNA-SWNTs than other ssDNA-SWNTs, in which the limit of detection reached 0.13% saturated vapor of P-ethylphenol. However, s6DNA-SWNTs can still be interfered with by other VOCs emitted by the strawberries in the view of poor selectivity. The bioelectronic nose took advantage of the different sensitivities of different gas biosensors to different VOCs. To improve measure precision, all ssDNA-SWNTs as a gas biosensor array were applied to monitor the different VOCs released by the strawberries, and the detecting data were processed by neural network fitting (NNF) and Gaussian process regression (GPR) with high accuracy.

## 1. Introduction

Electronic or bioelectronic noses are securing tremendous applications in food and environment safety, clinical diagnosis, and anti-bioterrorism [1,2], which mainly consists of a gas sensor/biosensor array and signal processing model. The development of nanotechnology has created vast potential to build highly sensitive, low-cost, portable electronic or bioelectronic noses with low power consumption [3,4]. Owing to the excellent electrical, optical, and mechanical properties, carbon materials, including fullerene, carbon black, carbon nanofiber, carbon nanotubes, and graphene, are ideal materials for developing the new generation of miniaturized, low-power, ubiquitous electronic or bioelectronic noses [5,6,7]. As one type of carbon nanotube, single-walled carbon nanotube (SWNT) is a one-atom-thick layer of graphite rolled up into a seamless cylinder with a diameter of several nanometers, and length on the order of 1–100 microns [8], which offers a high specific surface area to enhance the sensitivity. Based on electrical conductivity, there are two different types: semiconducting SWNT and metallic SWNT. The semiconducting SWNT is a hole-doped semiconductor, where the conductance of the SWNTs is observed to decrease by three orders of magnitude under positive gate voltages [9]. Even though SWNT-based sensors have been applied in the fields of trace gas molecules, their drawbacks are low sensitivity and selectivity.

To enhance the performance of SWNT-based sensors towards gas molecules, the surface of SWNT can be decorated with inorganic molecules (CuO, ZnO, TiO_2_, SnO_2_) [10,11,12], organic macromolecules (polyaniline, porphyrin, polycyclic aromatic hydrocarbons) [13,14,15] or biomolecules. Compared to inorganic and organic molecules, there are little types of biomolecules applied to modify SWNT to measure gas molecules. In the recent years, different single-strand DNAs (ssDNA) as a naturally occurring polymer has been explored to decorate SWNT through π–π interaction [16] that can respond to the particular odorants, which provides a novel material to develop biomimetic smell sensors for the chemical compounds detection [17]. Previous studies have demonstrated that the ssDNA decoration can change the response profiles of SWNTs to the volatile compounds due to the difference of the base number and sequence [18].

*Phytophthora cactorum* (*P. cactorum*) is a common phytopathogenic fungus, which can cause leather rot or crown rot diseases [19]. This disease in strawberries was firstly reported in 1924 in southern states of the USA and then spread from Asia to Europe subsequently over the next few decades. If strawberry plants are infected by *P. cactorum*, fruit losses of up to 20–30% or even 50% were reported according to the statistics [20]. There is an increasing need for early diagnosis of the *P. cactorum* infection to enable timely action to cut down fruit losses. Early diagnosis of *P. cactorum* infection in strawberry can be realized through direct molecular methods such as polymerase chain reaction (PCR) [21], illumination fluorescence (IF), fluorescence in situ hybridization (FISH) [22], electronic tongue (ET), and enzyme-linked immunosorbent assay (ELISA) [23]. Although the low limit of detection (LOD), low limit of quantification (LOQ), and high specificity can be achieved through the mentioned techniques, they are usually used for confirming the diseases after visual symptoms appear. However, due to the low concentration and non-uniform distribution of fungi in the tissues of infected strawberries, compounded by complex sample preparation and long assay times, these techniques are time-consuming, labor-intensive, and hence impractical for rapid detection of *P. cactorum* required in case of an epidemic. Jelen and coworker [20] was studied that volatile compounds present in strawberries infected with *P. cactorum*. Among the hundreds of volatile organic compounds (VOCs), two compounds were found to be responsible for the characteristic off-odour of strawberries infected by *P. cactorum*: 4-ethyl phenol (vapor pressure: 0.13 mmHg at 20 °C) and 4-ethyl-2-metoxyphenol (4-ethyl guaiacol) (vapor pressure: 0.017000 mmHg at 25 °C). The amount of these two compounds emitted by the infected strawberries ranged from 1.12 to 22.56 mg/kg and 0.14 to 1.05 mg/kg, respectively [24]. In contrast with 4-ethyl guaiacol, the amount of gas molecules of 4-ethyl phenol is much higher that can be responsible for the characteristic off-odor of *P. cactorum* infection. Therefore, early detection of *P. cactorum* in strawberries could be achieved through the volatile organic compound (VOC) of 4-ethyl phenol.

The odors of the infected plants or fruits are measured using spectrum based analytical instrument. The current state of the art techniques to assess the quality of products through odor evaluation and identification typically utilize gas chromatograph (GC) in combination with various detectors including flame ionization (GC-FID) [25], differential mobility spectrometry (GC-DMS) [26], headspace solid-phase microextraction (HS-SPME), high-performance liquid chromatography (HPLC), or mass spectrometry (GC-MS) [27]. They are not used for early detection, as these techniques require a laboratory setup, expensive instrumentation, and skilled personnel. However, these tools could not be used without a skilled analyst and are often unsuitable for in-field tests.

Electronic nose based on electrochemical gas sensor, which can be made portable, easy-to-use, and inexpensive, provides a solution to the shortages of spectrum devices. In this study, we propose a (bio)electronic nose using ssDNA decorated SWNTs coupling with a FET device to improve the sensitivity of bare SWNTs, which was applied to determine 4-ethyl phenol, the emissions of which are altered by the *P. cactorum* infected strawberry. Due to its non-invasive nature, it is amenable for in-field use when compared to other methods. To enhance the accuracy, the detecting data recorded by the ssDNA-SWNTs was processed using neural network fitting (NNF) and Gaussian process regression (GPR).

## 2. Materials and Methods

### 2.1. Chemicals and Reagents

Six different types of single-stranded DNA (ssDNA) [28,29] were synthesized and purified using standard solid-phase techniques, which were bought from Beijing Sunbiotech Co., Ltd. (Beijing, China). The ssDNA was dissolved in the sterilized Milli-Q water (100 μmol/L) and stored in refrigerator at −4 °C. The ssDNA sequences were listed below:s1-DNA: 5′-CTT CTG TCT TGA TGT TTG TCA AAC-3′s2-DNA: 5′-AAA ACC CCC GGG GTT TTT TTT TTT-3′s3-DNA: 5′-TTT TTT TTT TTT TTT-3′s4-DNA: 5′-GAG TCT GTG GAG GAG GTA GTC-3′s5-DNA: 5′-GTT GTT GGT TGT TG-3′s6-DNA: 5′-GTG TGT GTG TGT GTG TGT GTG TGT-3′

The semiconducting single-walled carbon nanotubes (SWNTs) were purchased from Nano-Integris Inc. (Beijing, China), which was homogeneously dispersed in Milli-Q water with a concentration of 0.01 mg/mL. Acetone, propanol, and ammonium hydroxide were acquired from Fisher Scientific Company (Pennsylvania, CP, USA). 3-Aminopropyltriethoxysilane (APTES), 2-pentanone, 2-heptanone, methyl butanoate, ethanol, methyl hexanoate, and ethyl butanoate were provided by Sigma-Aldrich (Beijing, China). Milli-Q water was used throughout the experiments. 2-Heptanol was offered by Acros Organics (Beijing, China). 4-Ethyl phenol was obtained from Sigma-Aldrich (Beijing, China). The All chemical reagents were analytical purity without purification. All the solutions were prepared using Milli-Q water (Billerica, MA, USA) with an electric resistance of 18.2 MΩ, which was further treated through high-temperature sterilization.

### 2.2. Apparatus

The micro-topography of SWNTs before and after non-covalently functionalized with ssDNA was investigated by Raman spectra (Thermo Scientific DXR 2xi, Shanghai, China), ultraviolet-visible spectra (UV–VIS; UV-1700, Shimadzu, Japan), scanning electron microscope (SEM; JEOL-6460, JSM-750, Beijing, China), and atomic force microscope (AFM; Bruker Bioscope Resolve, Massachusetts, MA, USA). Raman spectra were recorded using Dior XY Laser Raman (514 nm Diode and Ar ion lasers). The UV spectrum was collected by Beckman DU800 UV–VIS spectrophotometer (Beckman Coulter, Inc., Brea, CA, USA). SEM images were obtained by a Zeiss Leo SUPRA 55 with a beam energy of 10 kV. AFM images were taken using a Veeco Innova AFM. The electrochemical measurements, including current-voltage (*I*–*V*), field-effect transistor (FET), and current-time (I–T), were analyzed by a semiconductor parameter analyzer (Keithley 2636, Beijing, China).

### 2.3. Fabrication of Biosensor Arrays

The single gap electrode was fabricated on a high doped p-type silicon wafer, which covered with a 100 nm thick thermal oxide (SiO_2_). The wafer surface was spanned with photoresist and then written the single gap structure using standard lithographic patterning. After baking, a 20 nm Cr layer and a 180 nm Au layer were deposited on the surface via e-beam evaporation. Finally, the deposited wafer was immersed in acetone overnight to remove the unwritten parts, in which the size of the single gap was 10 μm wide by 10 μm long in Figure 1.

SWNTs based biosensor arrays were fabricated using the protocol as follows. The single gap electrode was cleaned sequentially with acetone and isopropanol, and then incubated in ammonium hydroxide for 30 min, which can eliminate the organic and inorganic substances on the surface. After that, the electrode was immersed in 0.5 mL APTES for 30 min, washed with Milli-Q water, and blow-dried with nitrogen. Moreover, 50 μL SWNTs solution was covered on the area between the source and drain with high humidity conditions in the dark for 60 min, followed by rinsing the SWNTs residue. The rinsed electrodes were annealed in CVD at 250 °C. Finally, SWNTs were functionalized with ssDNA by coating a 10 μL ssDNA solution on the electrode’s surface for 4 h under high humidity condition, and the residue was cleaned using Milli-Q water.

### 2.4. Gas Generator

The gas generator in Figure 2 was designed and integrated by our group, which was consisted of two mass flow controllers (MFCs), one bubbler, one air cylinder, and control software developed using LabView. The inlet ports of two MFCs were connected to the air cylinder directly, which were controlled by the control software. One of MFC was linked with the bubbler to generate the saturated vapor of VOC, and the other was employed to control the flow rate of dry air. Different concentrations of VOCs were achieved by mixing the different ratio of saturated VOCs and dry air. The sensing area of microelectrode was covered with a 1.2 cm^3^ sealed glass dome, which the gas molecules can pass through the glass dome.

### 2.5. Sensing Protocol

The relative resistance of ssDNA-SWNTs was defined as the following equation:(1)$$\frac{\Delta R}{R}=\frac{R-R_{0}}{R_{0}}\times 100\%$$
where *R*_0_ was the original resistance of ssDNA-SWNTs exposed to dry air, and *R* was the resistance of ssDNA-SWNTs exposed to VOCs.

## 3. Results and Discussion

### 3.1. The Characteristic of ssDNA-SWNTs

The characteristics of SWNTs before and after functionalized with ssDNA were investigated by different spectrum technologies, such as SEM, AFM, Raman, and UV–VIS.

Figure 3 shows the SEM images of bare SWNTs and ssDNA-SWNTs. It was clear that bare SWNTs were immobilized uniformly on the area of the microelectrode, which exhibited a net structure. After SWNTs modified with ssDNA, the image of ssDNA-SWNTs became blurred, and the diameter was slightly larger than the bare SWNTs. Due to the poor electrical conductivity of ssDNA, it indicated that ssDNA existed on the SWNTs’ surface.

The line-scan profiles of bare SWNTs before and after functionalized with ssDNA were analyzed by AFM, which was shown in Figure 4. The height of the bare SWNT was about 1.7 nm, which matched with the diameter of SWNTs reported by the vendor. The height of ssDNA-SWNT reached ~4 nm, demonstrating that ssDNA had been functionalized on the SWNT surface to form a uniform layer.

Raman spectra of bare SWNTs and ssDNA-SWNTs were shown in Figure 5. There were several peaks existed on the bare SWNTs, the D band, G band, and G’ band, which were located at 178 cm^−1^, 1345 cm^−1^, 1592 cm^−1^, and 2682 cm^−1^. After ssDNA modified on the SWNTs, the relative intensity of ssDNA-SWNTs was decreased greatly (The relative intensity of G^−^ to G^+^ band was reduced 9% relative to bare SWNTs [30]), and the G^+^ band was downshifted by 8 cm^−1^. The phenomenon might be attributed to ssDNA wrapping on SWNTs’ surface that blocked the SWNTs to contact with Raman or changed the charge transfer between the ssDNA and SWNTs.

To confirm the formation of ssDNA-SWNTs using the UV–VIS spectra, the SiO_2_/Si substrate was substituted by a quartz plate due to the poor optical transmittance. The UV absorption spectra of blank quartz, SWNTs/quartz, and ssDNA-SWNTs/quartz were shown in Figure 6. The spectrum of blank quartz was set as the background so that no peaks appeared in the entire wavelength region. When SWNTs were immobilized on the quartz’ surface, an absorption peak was found in the range from 200 nm to 300 nm. For the ssDNA-SWNTs/quartz, the absorbance peak located at 260 nm, but was much higher than SWNTs/quartz. The reason was that the characteristic absorption peak of SWNTs and ssDNA was located at 260 nm that can generate the composite effect on ssDNA-SWNTs/quartz [31,32].

Figure 7 shows the *I_DS_-V_DS_* of bare SWNTs before and after decorated with ssDNA. In the Figure 7A, *I_DS_* exhibited a linear relationship with the voltage ranging from −0.1 V to +0.1 V, and the slope of the *I_DS_-V_DS_* curve was decreased after ssDNA modification. The resistance of the ssDNA-SWNTs increased ~10.5-fold from 8.5 kΩ to 94.0 kΩ compared to bare SWNTs, resulting in a decrease of the charge carrier concentration of the p-type SWNTs and electron scattering from these molecules [33]. The ssDNA-SWNTs was further confirmed by obtaining FET transfer characteristics in Figure 7B. The threshold gate voltage (*V_TH_*) of bare SWNTs was 4.34 V. After SWNTs immobilized with ssDNA, the transfer curve shifted to the negative direction that *V_TH_* reached −6.28 V. Additionally, the mobility decreased from 140.51 cm^2^/Vs for bare SWNTs to 41.03 cm^2^/Vs for ssDNA-SWNTs. The shift in threshold voltage and a decrease in hole mobility can be explained by the reduction in hole concentration of p-type SWNTs due to charge (electron) transfer from the negatively-charged phosphate backbone of ssDNA.

### 3.2. Optimization

To simplify the experimental conditions, some parameters were optimized by detecting different concentrations of 4-ethyl phenol.

Figure 8 shows the relative resistance after different ssDNA-SWNTs exposed to the 100% saturated vapor of 4-ethyl phenol for 5 min. The relative resistance of bare SWNTs was about 1.26. However, SWNTs functionalized with different ssDNA displayed a decrease in resistance, so that the relative resistances showed negative values. There were only three absolute values of relative resistance of ssDNA-SWNTs were higher than bare SWNTs. Further, the sensitivity of the six gas biosensors were in the order of s6DNA-SWNTs > s4DNA-SWNTs > s1DNA-SWNTs > s2DNA-SWNTs > s5DNA-SWNTs > s3DNA-SWNTs, which was ascribed to the types and numbers of bases in different ssDNA. Different ssDNA functionalized with SWNTs might be formed different complex, sequence-specific set of binding pockets that had different combining abilities with 4-ethyl phenol [34]. The relative resistance of s6DNA-SWNTs reached −14.51, which was chosen to monitor the level of 4-ethyl phenol.

Figure 9 shows the relative resistances of s6DNA-SWNTs exposed to 5%, 10%, 40%, and 100% saturated vapor of 4-ethyl phenol for 7 min, and the resistances were recorded every 1 min. It was obvious that the relative resistance of s6DNA-SWNTs was positively related to the exposure time, especially for the low concentrations of 4-ethyl phenol. When the relative resistance of s6DNA-SWNTs reached the maximum upper the 10% saturated vapor of 4-ethyl phenol, the exposure time needed 4 min, 3 min, and 2 min for 10%, 40%, and 100% saturated vapor of 4-ethyl phenol, respectively. The gas molecules were absorbed by the gas sensing material on the s6DNA-SWNTs that will achieve a dynamic balance. The higher was concentration of 4-ethyl phenol, the shorter was its equilibrium time. Thus, 5 min was selected to identify the lower concentration of 4-ethyl phenol with high sensitivity.

The detecting voltage between the source and the drain (*V_DS_*) was one of the most important parameters for the bioelectronic nose. According to the voltage-current curves shown in Figure 7A, the current increased with voltage in the range from −0.1 V to 0.1 V, which displayed an excellent linearity relationship. The resistance at each voltage was calculated using the Ohm’s law, which was shown in Figure 10. It was clear that the values of bare SWNTs and s6DNA-SWNTs were almost equal at different voltage, excluding 0 V. The discontinuities at 0 V was mainly because the insufficient precision of device. The device offered an ultralow voltage, but cannot measure the ultralow current with high accuracy. To obtain a higher current signal of bioelectronic noise, 0.1 V was chosen as the detecting voltage.

### 3.3. Sensitivity for 4-ethyl Phenol Detection

Under the optimized parameters discussed above, the analytical performances of bare SWNTs and s6DNA-SWNTs were studied through monitoring different concentrations of the saturated vapor of 4-ethyl phenol. Figure 11A shows the dynamic response of bare SWNTs and s6DNA-SWNTs at *V_DS_* = 0.1 V and *V_G_* = 0 V, which were exposed to 4-ethyl phenol for 5 min and air for 10 min in proper order. The concentrations of the saturated vapor of 4-ethyl phenol were ranging from 0.25% to 100%. Compared with bare SWNTs without the s6DNA interlayer, the sensitivity of s6DNA-SWNTs was nearly one order of magnitude higher, indicating that s6DNA was a promising functional material for the detection of 4-ethyl phenol. Based on calibration plots shown in Figure 7B, the relative resistance increased gradually in the low concentration range and continued to rise sharply in the high concentration range. The relative resistance of s6DNA-SWNTs revealed an excellent relationship with the 4-ethyl phenol concentration in the range of 0.25% to 20% and 20% to 100%. The regression equation was $y_{1}=-0.5335x-0.871$ and $y_{2}=-0.0892x-5.6705$ with the linear regression correlation coefficient of 0.993 and 0.990, and the limit of detection was 0.13 (S/N = 3).

### 3.4. Repeatability

To study the repeatability, four different s6DNA-SWNTs were fabricated using the protocol mentioned above. These four s6DNA-SWNTs were used to determine the different concentrations of the saturated vapor of 4-ethyl phenol. Figure 12 shows the dynamic responses of four different s6DNA-SWNTs exposed to different concentrations of 4-ethyl phenol vapor for 5 min and flow air for 10 min at intervals. It was clear that the relative resistance was positive correlation with the 4-ethyl phenol concentration, and the trend of four curves was consistent. The relative standard deviation (RSD) at each concentration was lower than 7%, indicating that the s6DNA-SWNTs had excellent repeatability.

### 3.5. Sensing Mechanism

To identify the sensing mechanism, s6DNA-SWNTs were operated in a field-effect transistor mode using Si as the back-gate electrode. The changes in the transfer characteristic curves (*I_DS_-V_G_*, *V_DS_* = 0.1 V) upon functionalization with ssDNA and exposure to the VOCs were used to explain the electrostatic interactions of the ssDNA and VOC molecules with the SWNTs. Figure 13 shows transfer characteristics (*I_DS_-V_G_*) curves for s6DNA-SWNTs exposed to air and saturated vapor of 4-ethyl phenol. The transconductance (slope of the transfer characteristics curve) increased slightly after exposure to saturated vapor of 4-ethyl phenol, and the threshold gate voltage (*V_TH_*) also decreased to −11.02 V. This negative shift of *V_TH_* is attributed to adsorption of partially charged/polar VOC molecules that induce a screening charge (doping) of the SWNTs and shift the *I_DS_-V_G_* curve to a negative voltage. The mobility of s6DNA-SWNTs increased to 46.16 cm^2^/Vs in comparison with bare SWNTs. This mechanism was the electrostatic gating [35].

### 3.6. Interference

In multiple real-world environments, there were over 350 volatile compounds emitted by the strawberry. According to the previous research, the average relative proportion of 14 VOCs (ethyl pentanoate, methyl 3-methyl butanoate, methyl pentanoate, pentyl acetate, ethyl 3-methyl butanoate, 1-methyl ethyl butanoate acetaldehyde, 2-pentanone, 2-heptanone, methyl butanoate, ethanol, acetone, methyl hexanoate, ethyl butanoate) accounted for more than 82%. In comparison, the average relative proportion of the first seven VOCs was much lower than the last seven VOCs, which was only about 3.28%. The last seven VOCs (2-pentanone, 2-heptanone, methyl butanoate, ethanol, acetone, methyl hexanoate, ethyl butanoate) were chosen to evaluate the selectivity of s6DNA-SWNTs. Under the optimized conditions, Figure 14 shows the relative resistance of s6DNA-SWNTs were exposed to different saturated vapors, respectively. Only methyl hexanoate had no interference and s6DNA-SWNTs was difficult to discriminate 4-ethyl phenol from a mixture of VOCs.

### 3.7. Chemometric Analysis

To improve the accuracy of the measurement towards 4-ethyl phenol, the seven gas biosensors (s1DNA-SWNTs, s2DNA-SWNTs, s3DNA-SWNTs, s4DNA-SWNTs, s5DNA-SWNTs, s6DNA-SWNTs, bare SWNTs) were employed to the nine VOCs (methyl hexanoate, methyl butanoate, ethyl butanoate, ethanol, acetone, 2-pentanone, 2-heptanone, water, and 4-ethyl phenol) released by the infected strawberry, which the relative resistances at each concentration were shown in Figure 15. The dynamic responses for first eight VOCs were displayed in Appendix A.

#### 3.7.1. Neural Network Fitting

Neural network fitting (NNF) had been found useful and efficient, particularly in problems for which the characteristics of the processes are difficult to describe using physical equations. As the overall structure is shown in Figure 9, the procedure of NNF is mainly divided into three parts containing an input layer, hidden layers, and an output layer. In my model, the X matrix is work as the input layer, which consists of 100 columns and seven rows by using different concentrations (corresponding to 0.25%, 0.5%, 1%, 2.5%, 5%, 10%, 20%, 40%, 60%, 80%, and 100% saturated vapors) of nine VOCs as the columns and ssDNA functionalized SWNTs (s1DNA-SWNTs, s2DNA-SWNTs, s3DNA-SWNTs, s4DNA-SWNTs, s5DNA-SWNTs, s6DNA-SWNTs, bare SWNTs) as rows. The Y matrix is selected as an output layer, which consists of 100 columns and nine rows that four concentrations of each row corresponding to each column of X. The data of the X matrix is divided randomly into three sets: 70% as training samples, 15% of validation samples, and 15% of testing samples. During learning, output values from the NNF are compared to true values, and the coupling weights are adjusted to give a minimum sum of square errors. After testing and comparing, we found that ten nodes of the hidden layer can make the average relative error minimum shown in Figure 16. After the model was established, the predicted values from the NNF was compared to the true values, which were exhibited in Figure 17.

#### 3.7.2. Gaussian Process Regression

Gaussian process regression (GPR) was used to build the prediction model. The X matrix consisted of 100 rows, and seven columns are work as the workspace variable, and the Y matrix consisted of 100 rows and one column is selected as the response. After the model was established, the predicted values from the GPR are compared to the true values, which were exhibited in Figure 18.

NNF and GPR were simple methods that can offer qualitative and quantitative information, which widely employed in stoichiometry. The correlation coefficient (R_0_) and root mean square error (R_SMT_) between real concentrations and predicted concentrations were shown in Table 1. We found that the best results were achieved with the utilization of the GPR model because the R_0_ for 4-ethyl phenol was higher, and the R_SMT_ was lower in comparison with NNF.

## 4. Conclusion

A novel bioelectronic nose based on a gas biosensor array was fabricated using FET immobilized with SWNTs and ssDNA successively, and the detecting data was processed though the neural network fitting (NNF) and Gaussian process regression (GPR). The bioelectronic nose was used to measure the content of 4-ethyl phenol over a wide range of 0.25% to 100%, which was further diagnosed whether *P. cactorum* infected in strawberries or not. The ssDNA-SWNTs shows a significant improvement in sensitivity compared to bare SWNTs with excellent recovery and respectability.

## Figures and Tables

**Figure 1 nanomaterials-10-00479-f001:**
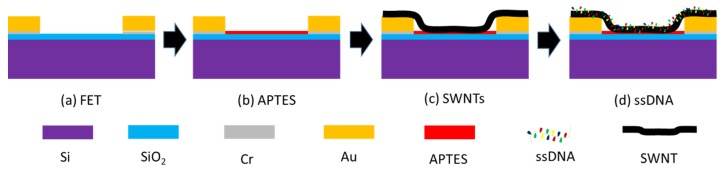
Single gap electrode decorated with SWNTs and ssDNA: (**a**) FET, (**b**) FET incubated in APTES, (**c**) SWNTs modified on the surface of FET and (**d**) ssDNA decorated on SWNTs.

**Figure 2 nanomaterials-10-00479-f002:**
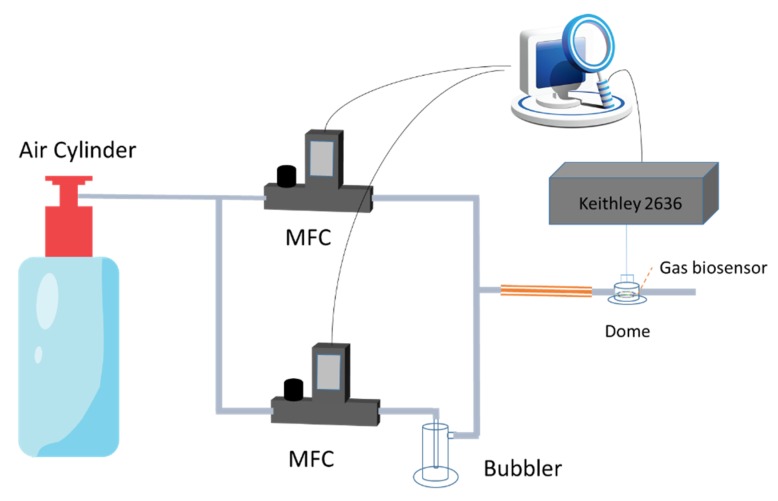
The schematic diagram of the gas generator.

**Figure 3 nanomaterials-10-00479-f003:**
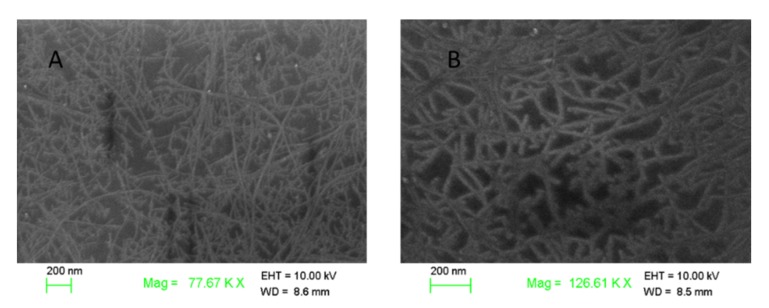
SEM images of (**A**) bare SWNTs and (**B**) ssDNA-SWNTs.

**Figure 4 nanomaterials-10-00479-f004:**
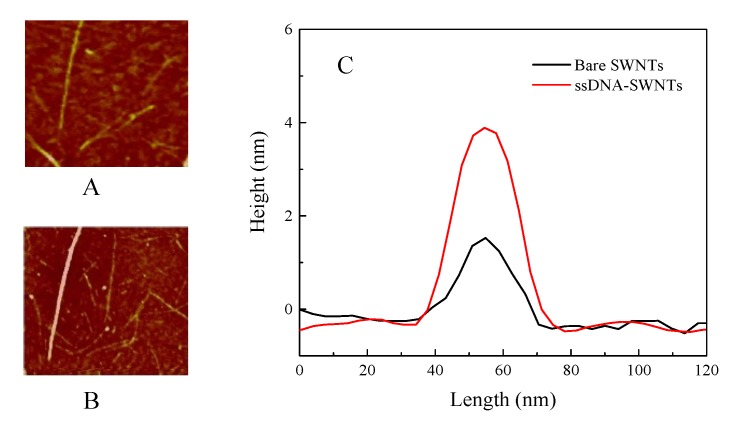
AFM image of (**A**) bare SWNTs; (**B**) ssDNA-SWNTs; (**C**) the height profile of bare SWNTs and ssDNA-SWNT.

**Figure 5 nanomaterials-10-00479-f005:**
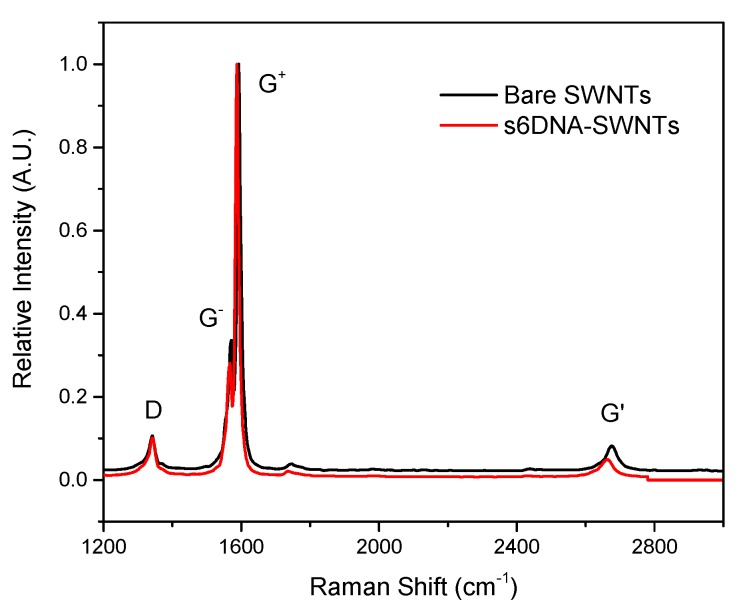
Raman spectrum of bare SWNTs (Black) and ssDNA-SWNTs (Red).

**Figure 6 nanomaterials-10-00479-f006:**
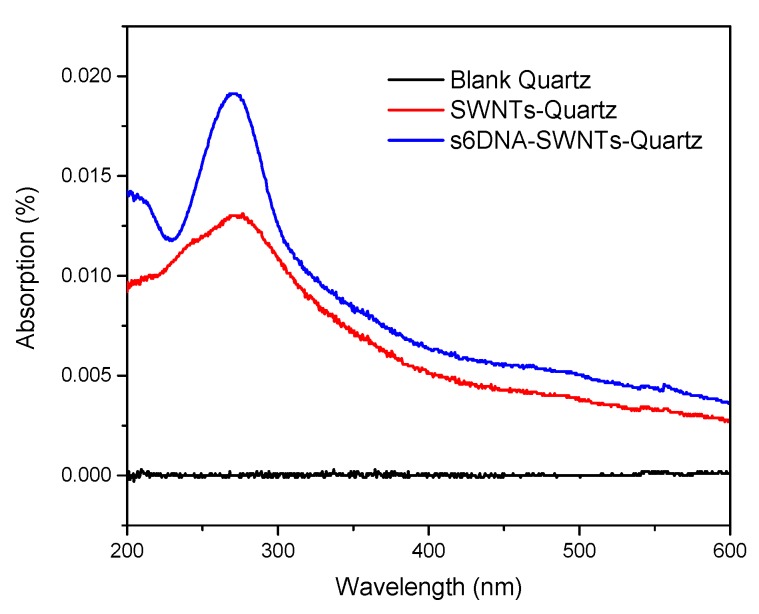
The UV spectrum of blank quartz (Black), SWNTs-quartz (Red) and s6DNA-SWNTs-quartz (Blue).

**Figure 7 nanomaterials-10-00479-f007:**
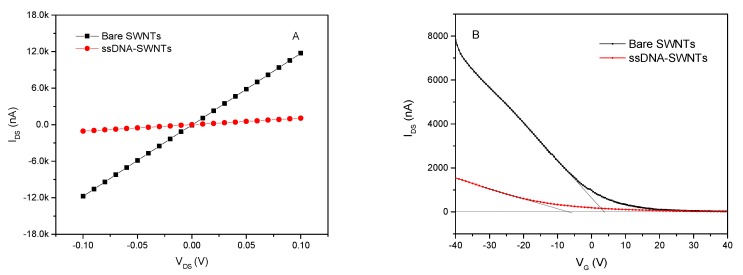
(**A**) *I_DS_-V_DS_* characteristics of bare SWNTs before and after functionalization with ssDNA at *V_G_* = 0 V; (**B**) transfer characteristics curve for bare SWNTs and ssDNA-SWNTs at *V_DS_* = 0.1 V and *V_G_* ranging from −40 V to +40 V.

**Figure 8 nanomaterials-10-00479-f008:**
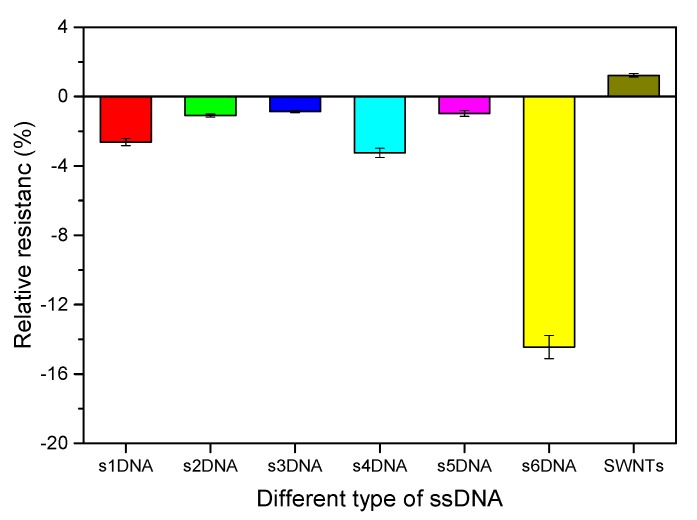
The relative resistance of SWNTs before and after functionalized with different ssDNA exposed to the saturated vapor of 4-ethyl phenol for 5 min (*V_DS_* = 0.1 V and *V_G_* = 0 V; each point calculated by the average value of four different s6DNA-SWNTs).

**Figure 9 nanomaterials-10-00479-f009:**
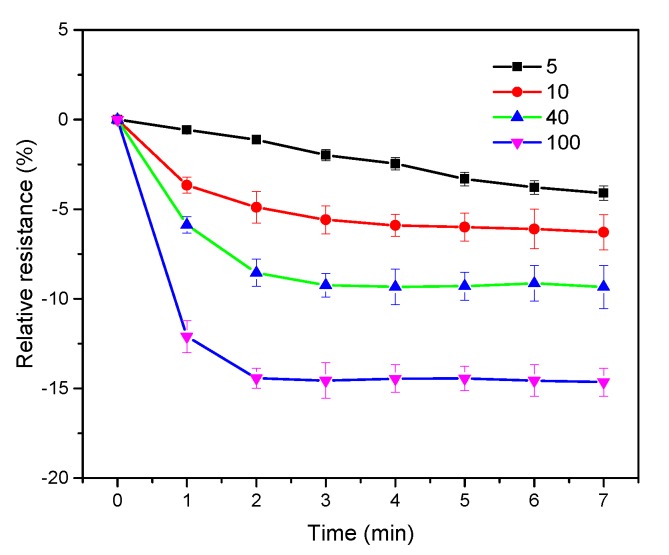
The relative resistance of s6DNA-SWNTs exposed to 5%, 10%, 40%, and 100% saturated vapor of 4-ethyl phenol for 7 min (*V_DS_* = 0.1 V and *V_G_* = 0 V; each point calculated by the average value of four different s6DNA-SWNTs).

**Figure 10 nanomaterials-10-00479-f010:**
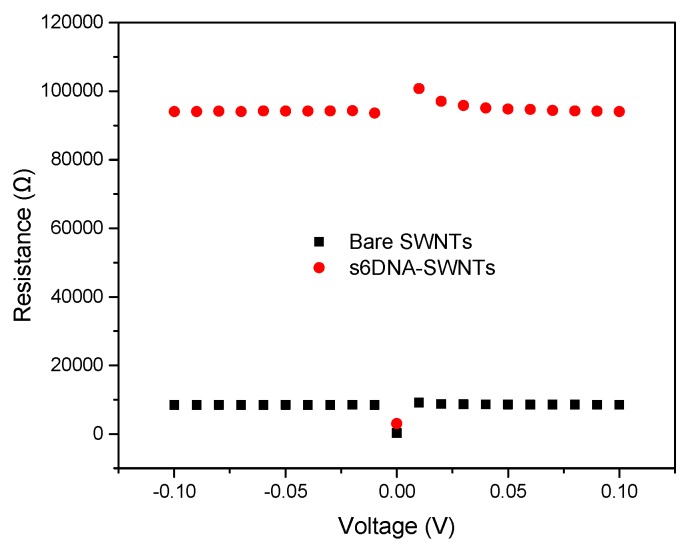
The resistance of SWNTs before and after functionalized with different s6DNA at different voltages.

**Figure 11 nanomaterials-10-00479-f011:**
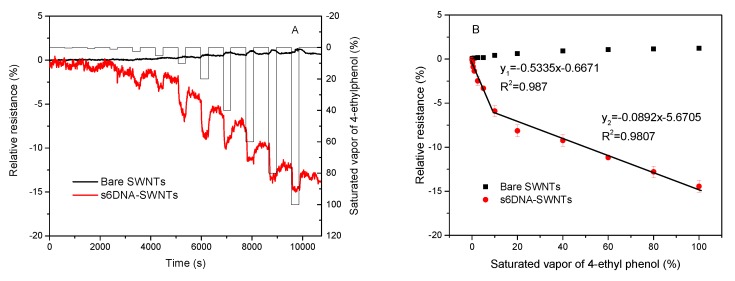
(**A**)The dynamic responses and (**B**) the relative resistances of bare SWNTs and s6DNA-SWNTs towards different concentrations (0%, 0.25%, 0.5%, 1.0%, 2.5%, 5%, 10%, 20%, 40, 60%, 80%, and 100%) of the saturated vapor of 4-ethyl phenol (*V_DS_* = 0.1V and *V_G_* = 0 V; each point calculated by the average value of four different gas biosensors).

**Figure 12 nanomaterials-10-00479-f012:**
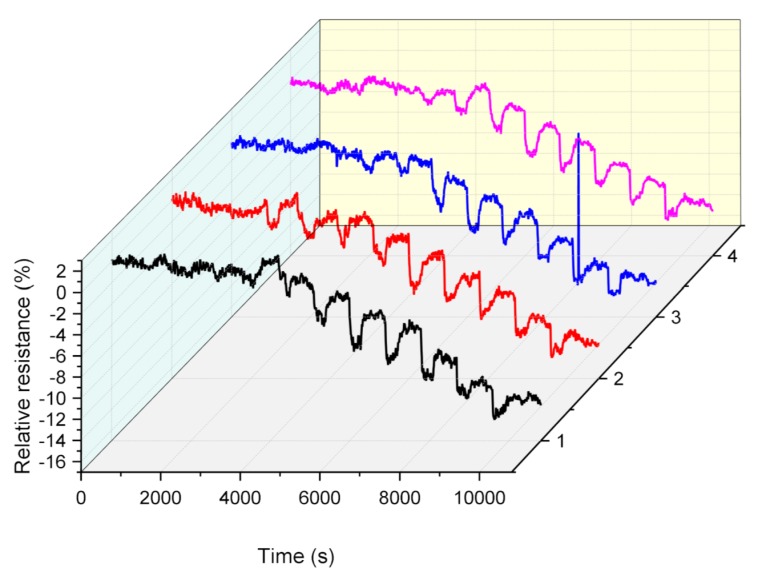
The dynamic responses of four different s6DNA-SWNTs exposed to different concentrations (0, 0.25, 0.5, 1.0, 2.5, 5, 10, 20, 40, 60, 80 and 100) of 4-ethyl phenol vapor for 5 min and flow air for 10 min at intervals. (*V_DS_* = 0.1 V and *V_G_* = 0 V; each concentration exposed for 5 min and point calculated by the average value of four different gas biosensors).

**Figure 13 nanomaterials-10-00479-f013:**
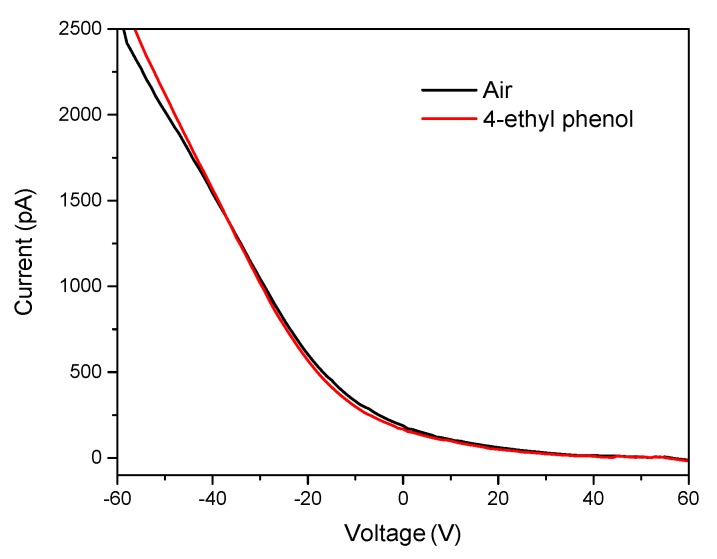
The transfer characteristics curve of s6DNA-SWNTs exposed to air and saturated vapor of 4-ethyl phenol for 5 min. (*V_DS_* = 0.1 V and *V_G_* in the range from −60 to +60 V with the scan rate 1 V).

**Figure 14 nanomaterials-10-00479-f014:**
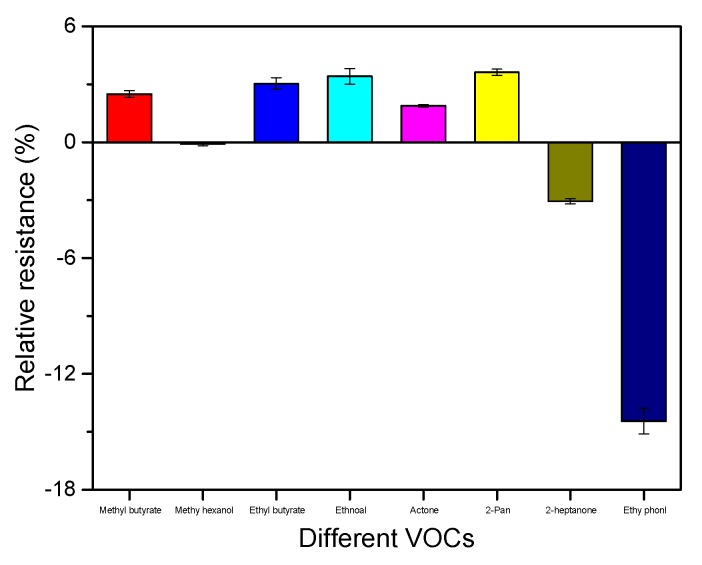
The relative resistances of s6DNA-SWNTs exposed to different saturated vapors of VOCs (methyl butanoate, methyl hexanoate, ethyl butanoate, ethanol, acetone, 2-pentanone, 2-heptanone, 4-ethyl phenol) for 5 min. (*V_DS_* = 0.1 V and *V_G_* = 0 V).

**Figure 15 nanomaterials-10-00479-f015:**
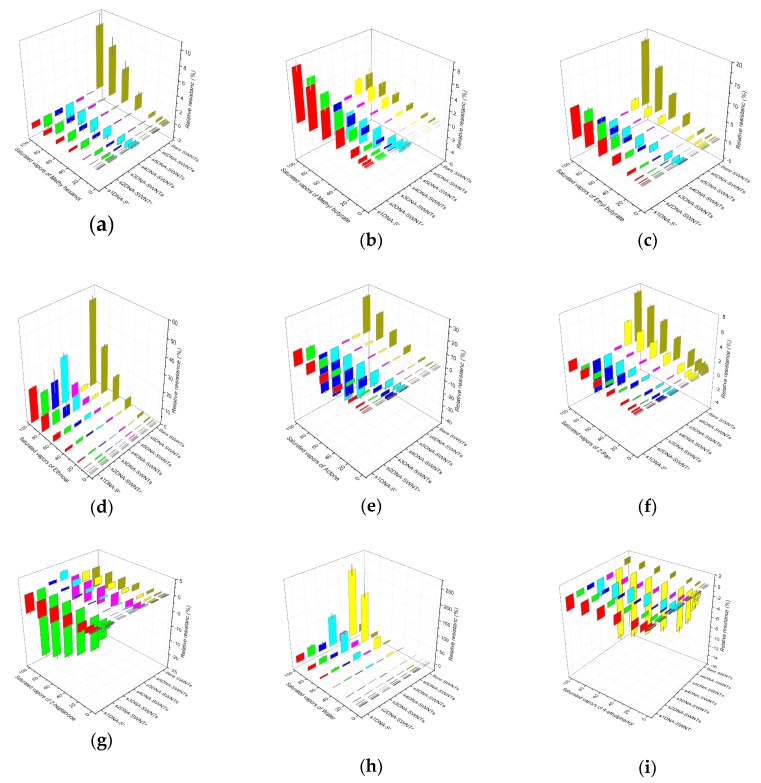
The relative resistance of different ssDNA-SWNTs (s1DNA-SWNTs, s2DNA-SWNTs, s3DNA-SWNTs, s4DNA-SWNTs, s5DNA-SWNTs, s6DNA-SWNTs, bare SWNTs) towards different concentrations of different VOCs (**a**–**i**: methyl hexanoate, methyl butanoate, ethyl butanoate, ethanol, acetone, 2-pentanone, 2-heptanone, water, and 4-ethyl phenol) varying from 0.25% to 100%. (*V_DS_* = 0.1 V and *V_G_* = 0 V, each point calculated by the average value of four different ssDNA-SWNTs).

**Figure 16 nanomaterials-10-00479-f016:**
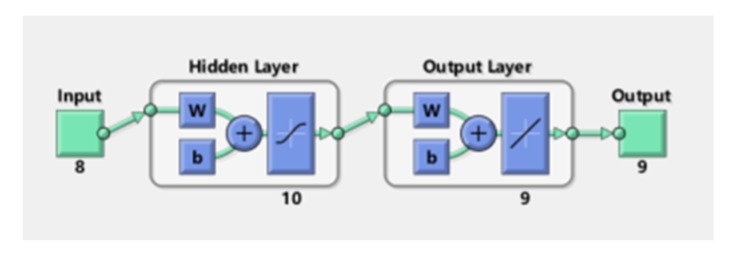
Type of architecture of the neural network fitting.

**Figure 17 nanomaterials-10-00479-f017:**
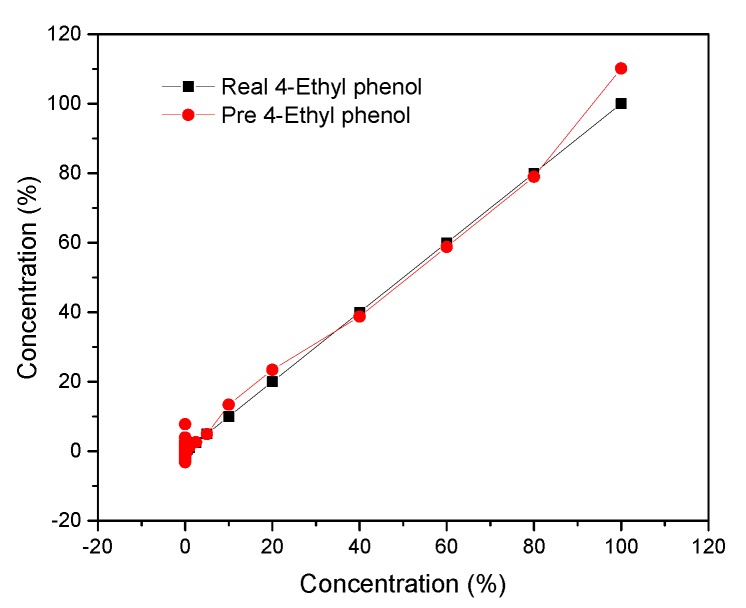
Predicted concentration against true concentration of 4-ethyl phenol for the NNF model.

**Figure 18 nanomaterials-10-00479-f018:**
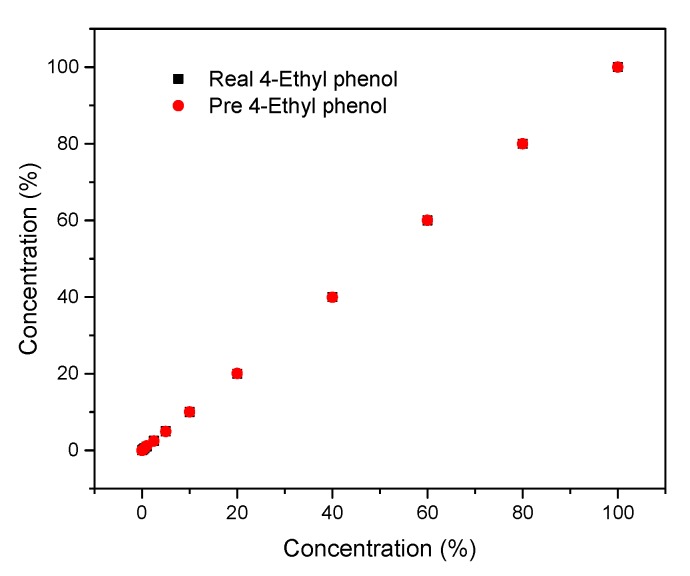
Predicted concentration against true concentration of 4-ethyl phenol for the GPR model.

**Table 1 nanomaterials-10-00479-t001:** The correlation coefficient (R_0_) and root mean square error (R_SMT_) between real concentrations and predicted concentrations.

	R_0_	R_SMT_
NNF	0.98321	4.82493
GPR	0.99883	1.33108

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
