# Peer review of "Bioelectronic Nose Based on Single-Stranded DNA and Single-Walled Carbon Nanotube to Identify a Major Plant Volatile Organic Compound (p-Ethylphenol) Released by Phytophthora Cactorum Infected Strawberries"

_nanomaterials, 2020, doi:10.3390/nano10030479_

Round 1

Reviewer 1 Report

In this manuscript, Hui Wan and coworkers developed bioelectronic nose based on a gas biosensor array and signal processing model is developed for the noninvasive diagnostics of the P. cactorum infected strawberries, which overcomes the limitations of the traditional spectral analysis methods. The gas biosensor array was fabricated using the single-wall carbon nanotubes (SWNTs) immobilized on the surface of field-effect transistor, and then non-covalently functionalized with different single-strand DNAs (ssDNA) through π- π interaction. The bioelectronic nose was used to measure the content of 4-ethyl phenol over a wide range of 0.25 to 100%, which was further diagnosed  whether P. cactorum infected in strawberries or not. The ssDNA-SWNTs shows a significant improvement in sensitivity compared to bare SWNTs with excellent recovery and respectability.

It is very important for new nanotechnology to detect P. cactorum because the P. cactorum are now attracting attention. However, there are points that are still not clear. This manuscript should be revised at least for the following:

1) The author wrote that this sensor performs molecular recognition by π-π interaction. However, there are concerns that compounds with a π-plane exist naturally and lack target selectivity. Authors should investigate and discuss in detail how the target molecule is selected

2) The author only showed experimental results and did not fully discuss why the phenomenon was caused. Authors should consider the results carefully.

3)The captions in the figures are not sufficient and difficult to understand. The authors should explain in more detail.

4)The description of the figure is not appropriate in the text. For example, the reviewer finds it difficult to understand where the description in Figure 12 is located in the text chapter.

Author Response

1) The author wrote that this sensor performs molecular recognition by π-π interaction. However, there are concerns that compounds with a π-plane exist naturally and lack target selectivity. Authors should investigate and discuss in detail how the target molecule is selected

Response: The six types of single-stranded DNA were selected according to the previous three articles.

Staii, Cristian, et al. "DNA-decorated carbon nanotubes for chemical sensing." Nano Letters 5.9 (2005): 1774-1778.

Kybert, Nicholas John. "Nano-Bio Hybrid Electronic Sensors for Chemical Detection and Disease Diagnostics." (2015).

Johnson, AT Charlie, et al. "DNA-decorated carbon nanotubes for chemical sensing." Semiconductor science and technology 21.11 (2006): S17.

SWNTs was modified with ssDNA though π-π interaction.  The sensing mechanism, why ssDNA-SWNTs are more sensitive to gas molecules, was discussed in the section 3.5. Now, we only can explain the phenomenon through the changes of semiconductor performance. Relationship between ssDNA structure and gas molecular is still under study that will explain the detail in the future.

2) The author only showed experimental results and did not fully discuss why the phenomenon was caused. Authors should consider the results carefully.

Response:  Thanks for your advices that parts of experimental results did not fully discuss. We have checked and revised this “Results and Discussion” section. The explanations are supplemented, which are marked using red color. Please check the manuscript!

3)The captions in the figures are not sufficient and difficult to understand. The authors should explain in more detail.

Response:  Thanks for your advices. The captions in the manuscript have been revised that make them understand easily.

4)The description of the figure is not appropriate in the text. For example, the reviewer finds it difficult to understand where the description in Figure 12 is located in the text chapter.

Response:  Thanks for your advices.

Figure 12 shows the dynamic responses of four different s6DNA-SWNTs exposed to different concentrations of 4-ethyl phenol vapor for 5 min and flow air for 10 min at intervals. It was clear that the relative resistance was positive correlation with the 4-ethyl phenol concentration, and the trend of four curves was consistent. The relative standard deviation (RSD) at each concentration was lower than 7 %, indicating that the s6DNA-SWNTs had excellent repeatability.

Reviewer 2 Report

The authors present efforts towards the development of a biosensor based on carbon nanotubes wrapped in DNA for volatile organics released from fungus infected strawberries. The approach is interesting and promising results are shown for detection of 4-ethylphenol, a major volatile released from the infected strawberries. One of 6 used ssDNA strands is found to have a good response to 4-ethylphenol in terms of resistance change. There are some points that could use improvement:

(a) while 6 different ssDNA strands are chosen, there is not much discussion of the rationale for why these 6 were chosen as far as either sequence, base pair composition or length or what is special about sequence #6

(b) the small colorful DNA in Figure 1 should be made a bit more visually evident in the last step of DNA decoration

(c)  on line 82, 4-ethyl-2-methoxyphenol is spelled wrong

(d) please label the actual sensor in Figure 2

(e) please add a sentence explaining the discontinuities near 0 V in Figure 10

Author Response

Thanks for your advice. All  questions have been answered in the attachment.  

Round 2

Reviewer 1 Report

The authors responded to the reviewer's comments and modified the manuscript. Thus, this manuscript is much improved.